# The carbonization of aromatic molecules with three-dimensional structures affords carbon materials with controlled pore sizes at the Ångstrom-level

Tomoki Ogoshi [1,2✉], Yuma Sakatsume[3], Katsuto Onishi[1], Rui Tang[4], Kazuma Takahashi[5],
Hirotomo Nishihara [4,5], Yuta Nishina [6], Benoît D. L. Campéon [6], Takahiro Kakuta[2,3] &
Tada-Aki Yamagishi[3]

Carbon materials with controlled pore sizes at the nanometer level have been obtained by template methods, chemical vapor desorption, and extraction of metals from carbides. However, to produce porous carbons with controlled pore sizes at the Ångstrom-level, syntheses that are simple, versatile, and reproducible are desired. Here, we report a synthetic method to prepare porous carbon materials with pore sizes that can be precisely controlled at the Ångstrom-level. Heating first induces thermal polymerization of selected three-dimensional aromatic molecules as the carbon sources, further heating results in extremely high carbonization yields (>86%). The porous carbon obtained from a tetra-biphenylmethane structure has a larger pore size (4.40 Å) than those from a spirobifluorene (4.07 Å) or a tetraphenylmethane precursor (4.05 Å). The porous carbon obtained from tetraphenylmethane is applied as an anode material for sodium-ion battery.

[1] Department of Synthetic Chemistry and Biological Chemistry, Graduate School of Engineering, Kyoto University, Kyoto, Japan. [2] WPI Nano Life Science Institute (WPI-NanoLSI), Kanazawa University, Kakuma-machi, Kanazawa, Japan. [3] Graduate School of Natural Science and Technology, Kanazawa University, Kakuma-machi, Kanazawa, Japan. [4] Advanced Institute for Materials Research (WPI-AIMR), Tohoku University, Sendai, Miyagi, Japan. [5] Institute of Multidisciplinary Research for Advanced Materials, Tohoku University, Sendai, Miyagi, Japan. [6] Research Core for Interdisciplinary Sciences, Okayama University, Okayama, Japan. ✉email: ogoshi@sbchem.kyoto-u.ac.jp

Since the late 1990s, the preparation of porous carbon materials with designed micro- and meso-pores has been widely investigated. These porous carbon materials have been applied as absorbents, catalyst supports, and electrode materials because they show superior mechanical and chemical stability, and electrical conductivity[1–13]. One promising approach to produce such porous carbon materials with controlled pore sizes is the template method[6,14]. First, organic carbon sources are loaded into inorganic templates to form nano-composites. Second, calcination of the nanocomposites in an inert atmosphere results in conversion of the organic carbon source to carbon. Finally, the inorganic templates are removed from the composites by chemical etching. After the etching, carbon materials with controlled pore sizes at the nanometer level are obtained as negative replicas of the inorganic templates. This method is useful to obtain carbon materials with controlled pores. However, pore-size control at the Ångstrom level has been difficult owing to the unavailability of an inorganic template that can homogeneously accommodate a sufficient amount of the carbon source in their pores. There are several other classical methods that have been used to make porous carbons, such as chemical vapor deposition on activated carbons[15]. However, in these methods, the maximum carbon yields after the carbonization are approximately 30%, which indicate that the original carbon precursor structures are completely destroyed during the carbonization process. The extraction of metals from carbides is a well-known technique to make porous carbons with a control of the pore size[16]. However, the process is costly and generates gases that are toxic and corrosive. Furthermore, these methods depend on a craftsmanship process, which is based on a number of empirical parameters. This craftsmanship process has been the greatest hinderance to obtain porous carbons with controlled pores at the Ångstrom level using the procedures reported by different research groups. Therefore, to produce porous carbons with controlled pores at Ångstrom level, methods that are simple, versatile, and reproducible are highly desired. The carbonization of molecular-based porous frameworks, such as porous coordination polymers (PCPs) or metal–organic frameworks (MOFs), and covalent organic frameworks (COFs) has been attempted[17–27]. However, in most cases, carbonization induces significant damage to the frameworks, which results in a structural change of the original pores. Previously, we successfully synthesized carbon materials with controlled pore sizes at the Ångstrom level by carbonization of assemblies of pillar[6]arenes[28]. Carbonization of a two-dimensional sheet assembly constructed from pillar[6]arenes, which are hexagonal prism-shaped macrocyclic compounds, yielded carbons having a pore size identical to the original pore size of the pillar[6]arene molecule. Subsequently, we developed advanced techniques for pore-size control in carbon materials by the molecular design of a precursor. Herein, we report the synthesis of carbon materials with pore-size control at the Ångstrom level by the molecular design of carbon sources (Fig. 1).

As observed from the carbonization of pillar[6]arene, a thermally stable three-dimensional (3D) structure is effective to retain the molecular-level structures. Additionally, it has been shown that the introduction of ethynyl[29,30] or diacetylene[31] into the precursor molecule induces 3D cross-linking simply by heating, and this allows for a high yield of carbonization of the precursor molecule.

Considering the above background, we selected the thermally stable 3D aromatics, tetraphenylmethane (1, Fig. 1) and tetrabiphenylmethane (2), which are functionalized with four thermally polymerizable ethynyl groups. The four ethynyl groups induce isotropic and rigid 3D cross-linking, thus the pore size can be controlled by the length of the phenyl or biphenyl moieties. The carbon material C1 prepared by carbonization of 1 had

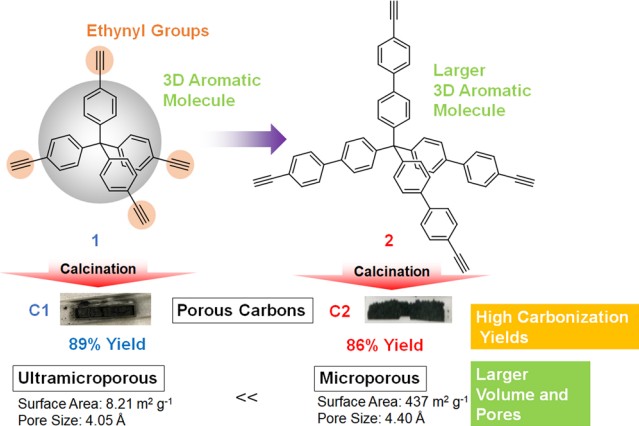

**Fig. 1 Molecular design of carbon sources.** Carbon sources (**1** and **2**) with three-dimensional (3D) aromatic molecules and thermally polymerizable ethynyl groups. Carbonization of these carbon sources afford carbon materials with controlled pore sizes at the Ångstrom level (**C1** and **C2**).

ultramicropores (4.05 Å). The carbon material **C2** from **2** had micropores (4.40 Å), which were larger than the pores of **C1**. Thus, by molecular design of the carbon sources, we were able to demonstrate the chemically sophisticated pore-size control of porous carbons at the Ångstrom level. The entire process of the present method involves the direct carbonization of the precursor molecules without the use of a template that would need to be removed by using toxic chemicals such as HF and NaOH. The carbonization takes place in the solid-state without significant emission of toxic gases. Furthermore, neither liquid tar nor any waste liquid caused by template removal are involved. Therefore, the proposed method is beneficial from the viewpoints of low environmental impact and sustainability. Interestingly, the porous carbon **C1** derived from **1** shows high conductivity and works as an anode electrode material for not only lithium- but also sodium-ion batteries.

## Results

**Synthesis of porous carbons.** Tetraphenylmethane with four ethynyl groups, **1**, was synthesized according to the previous work by W. Lu et al.[32]. Details of the synthetic procedure of the carbon source, tetrabiphenylmethane with four ethynyl groups, **2**, are provided in the Supporting Information. The thermal behavior of **1** and **2** was investigated by thermo-gravimetric analysis (TGA, Fig. 2a) and differential thermal analysis (DTA) in $N_2$.

The DTA curve of **1** exhibited an intense exothermic peak at 230 °C (blue dash line). An exothermic peak was also observed at 242 °C in the DTA curve of **2** (red dash line). As a reference, the thermal behavior of tetrakis(4-bromophenyl)methane (without ethynyl groups), **3**, was also investigated (Supplementary Fig. 6). The exothermic peak was not observed in the same temperature range for the DTA curve of **3**. Therefore, the exothermic peak corresponded to the thermal polymerization of ethynyl groups. Similar exothermic peaks from the thermal polymerization of organic compounds with ethynyl groups in the same temperature range have been reported[29,30]. After the polymerization, both polymerized samples adsorbed $CO_2$ and $N_2$ (Supplementary Figs. 7 and 8); thus the porous structures were developed by thermal polymerization. From TGA, **1** and **2**, after calcination at 900 °C, afforded carbon materials (**C1** and **C2**) in high yields of 89% and 86%, respectively (blue and red solid lines). In contrast, **3** showed a large weight loss at around 300 °C, and its carbonization yield after heating at 900 °C was only 2% (Supplementary Fig. 6). From these results, thermal

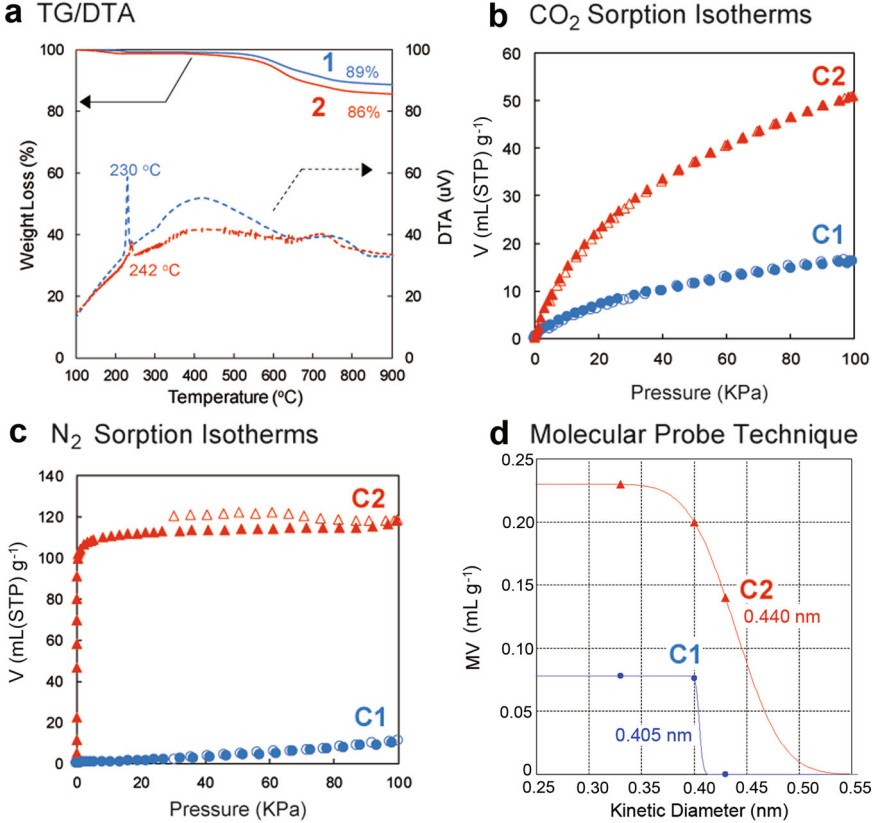

**Fig. 2 Carbonization and determination of pore sizes by gas sorption measurements. a** Thermo-gravimetric analysis (TGA, **1**: blue solid line, **2**: red solid line) and differential thermal analysis (DTA, **1**: blue dash line, **2**: red dash line) under a nitrogen atmosphere (heating rate: 1 °C / min). **b** $CO_2$ (25 °C) and **c** $N_2$ (−196 °C) sorption isotherms of the powders of porous carbons **C1** (blue circles) and **C2** (red triangles). Solid symbols = adsorption; open symbols = desorption. **d** Micropore volume (MV) of **C1** (blue circles) and **C2** (red triangles) estimated by the DA method, plotted against kinetic diameter of molecular probes.

polymerization of the ethynyl groups resulted in the formation of cross-linked polymers, which contributed to the high yield of production of carbon materials **C1** and **C2** by carbonization. The carbonization yields were quite high compared with previous examples of molecular precursors. For example, carbonization of ionic liquids afforded carbon materials with a high carbonization yield of approximately 22%[33]. Carbonization yield of pillar[6] arene assemblies was approximately 55%[28]. The present high carbonization yields of **1** and **2** are comparable to those of porphyrin derivatives with ethynyl groups[29,30].

**Characterization of porous carbons**. We investigated the carbon materials **C1** and **C2** obtained by calcination. The CHO ratios were precisely analyzed by high-sensitivity temperature-programmed desorption (TPD) up to 1800 °C, which enabled the detection of ppm-level H and O contents. The carbon contents in carbon materials **C1** and **C2** were approximately 98% (details in Supplementary Table 1), which indicated the formation of carbon-rich materials. The oxygen contents in carbon materials **C1** and **C2** were less than 2%. The oxygen contents of typical activated carbons are 3–8%, thus the oxygen contents of these carbon materials were relatively small compared with typical activated carbons. The origin of elemental oxygen resulted from oxidation of the dangling bonds after the carbonization. Generally, carbons after carbonization contain dangling bonds (radicals), which are easily oxidized upon exposure to air[31]. The Raman spectra of **C1** and **C2** showed carbon D- and G-bands at 1355 and 1590 cm$^{-1}$, respectively (Supplementary Fig. 9), which indicated the formation of a defective carbon structure. To

determine the crystallinity of **C1** and **C2**, the powder X-ray diffraction patterns were measured (Supplementary Figs. 10 and 11). The powders of **1** and **2** showed clear diffraction patterns, which indicated the crystallinity of **1** and **2**. After calcination, the sharp peaks disappeared, which indicated that the crystal structures of **1** and **2** became amorphous. At the same time, broad peaks were observed at around 22° and 45°, which corresponded to the carbon (002) and (10) planes, respectively, and are typical in nanoporous carbon materials.

**Pore size of porous carbons**. The structures of carbon materials **C1** and **C2** were observed by transmission electron microscopy (TEM), as shown in Fig. 3.

Unlike conventional activated carbons, graphene stacking structures were not found. Instead, numerous micropores with a surprisingly uniform size of less than 1 nm (white dots) were observed in both samples. Thus, the origin of the pores in carbon materials **C1** and **C2** was not the inter-sheet spaces called slit pores which are typical in general porous carbon materials, but spherical spaces derived from the developed 3D cross-linking of **1** and **2**. However, we were unable to observe a clear difference between the pore sizes of carbon materials **C1** (Fig. 3a) and **C2** (Fig. 3b) owing to the limitation of the TEM resolution. To investigate the porosity of the obtained carbon materials **C1** and **C2** in detail, we performed gas sorption measurements. Figure 2b shows the $CO_2$ (diameter: 3.3 Å) gas sorption isotherms of the powders of **C1** and **C2**. Both samples showed $CO_2$ uptake, which indicated that **C1** and **C2** had pores larger than $CO_2$ ( > 3.3 Å). **C2** (red triangles) adsorbed a much larger amount of $CO_2$ gas

**Fig. 3 TEM images.** TEM images of porous carbons of **a C1** and **b C2**.

than **C1** (blue circles), owing to the larger pore size and volume that were developed by the biphenyl moieties. When nitrogen gas was used (Fig. 2c), the powder of **C2** adsorbed $N_2$ (diameter: 3.7 Å, red triangles), but that of **C1** did not (blue circles). **C2** adsorbed ethane (diameter: 4.0 Å) and *n*-butane (diameter: 4.3 Å), but **C1** hardly adsorbed ethane and was unable to uptake *n*-butane gas (Supplementary Fig. 12). The pore sizes of **C1** and **C2**, which were determined by the molecular probe technique, were 4.05 and 4.40 Å, respectively (Fig. 2d). Therefore, we were able to successfully control the porosity of porous carbons by the molecular design of carbon sources. We also investigated the surface area by using $N_2$ gas and the Brunauer−Emmett−Teller (BET) method (Supplementary Fig. 13). The surface area of **C1** was much lower than that of **C2** (8.21 vs 437 $m^2$ $g^{-1}$). **C1** would not have continuous pores, thus $N_2$ molecules were unable to access the inner pore spaces of **C1** at −196 °C. In contrast, the size of the **C2** pore was larger than that of $N_2$ molecules, thus $N_2$ molecules were able to access the continuous inner pore spaces of **C2**. Therefore, the surface area of **C1** calculated by the $N_2$ molecule probes was too small compared with that of **C2**.

We prepared porous carbons from another aromatic precursor spirobifluorene with four acetylene groups (**4**). In this case, the obtained carbon adsorbed $CO_2$ and ethane (Supplementary Fig. 14), and had pores of 4.07 Å (Supplementary Fig. 15), which were a similar size to those of **C1** because the acetylene-modified spirobifluorene **4** has a 3D structure, that is similar to **1**. From the result, it was revealed that carbonization of 3D structures with acetylene groups is a useful way to prepare porous carbons with controlled pores at the Ångstrom level.

The thermal stability of these porous carbons is a very important point for practical applications. When heated again at 900 °C under nitrogen atmosphere (Supplementary Fig. 16a), the porous carbon **C2** which was synthesized by a simple template-free pyrolysis, did not show any loss of porosity. Such a stable behavior is comparable to that of conventional activated carbons such as MSC-30 (Supplementary Fig. 16b), and is superior compared with the high-performance materials synthesized by template techniques. For example, zeolite-templated carbon (ZTC), an ordered microporous carbon known as a high-performance material, was not stable at 900 °C. The surface area of ZTC was decreased by ca. 40% after heating (Supplementary Fig. 16c).

**Application of porous carbon materials for lithium- and sodium-ion batteries**. Porous carbons that can adsorb $CO_2$ gas and cannot adsorb $N_2$ gas are good candidates for anode active materials; thus, we used **C1** as an anode active material for a battery. When **C1** was used in a lithium-ion battery (LIB; Fig. 4a), **C1** showed a capacity comparable to that of graphite. The ionic volume of a lithium ion (1.84 $Å^3$) is small, and therefore graphite can intercalate and de-intercalate lithium ions, which is the

reason why graphite works as an anode active material. **C1** also has a porous structure, thus **C1** can absorb and desorb lithium ions (Supplementary Fig. 17), and works as an anode active material. Interestingly, when **C1** was used in a sodium-ion battery (SIB; Fig. 4b), **C1** exhibited a capacity approximately 1.7 times higher than that of graphite. The ionic volume of a sodium ion is 4.44 $Å^3$, which is larger than that of a lithium ion (1.84 $Å^3$). Owing to the large size of a sodium ion, graphite can hardly intercalate sodium ions, which results in a low capacity. However, **C1** has pores, thus larger sodium ions can move in the pores of **C1**, which results in the high capacity. We also used **C2** in lithium- and sodium-ion batteries (Supplementary Fig. 18). The capacity of **C2** was approximately 0.8 times smaller than that of **C1**. From the gas sorption experiments, the cavity size of **C2** was larger than that of **C1**. The cavity of **C2** would therefore be too large for the lithium and sodium cations. Therefore, the pore size control by molecular design of the carbon sources in this study is very important. For application of the porous carbon as an electrode material, the conductivity is an essential property. Therefore, we measured the conductivity of **C1** (Fig. 4c). **C1** showed good conductivity compared with Gramax (synthetic graphite material used for lithium-ion batteries) and YP50F (activated carbon used for electric double-layer capacitors).

## Discussion

We were able to prepare carbon materials with pore sizes controlled at the Ångstrom level by carbonization of the designed carbon sources, which were 3D aromatic molecules with ethynyl groups. Introducing ethynyl groups contributed to high carbonization yields because of the formation of thermally stable cross-linked polymers. Furthermore, the carbon materials had pore sizes controlled at the Ångstrom level. The pore size of the carbon materials was able to be increased by using large 3D aromatic molecules. To date, there have been few examples of controlling the pore size of carbon materials by molecular design of the carbon sources. The novelty in this study is the preparation of porous carbons resulting from molecular design of carbon precursors. In this method, we successfully obtained porous carbons with pore-size control at the Ångstrom level by simple carbonization of the carbon precursors. Owing to the high carbonization yields, the porous carbon structures reflect the structure of the original precursor, and Ångstrom level pore-size control based on the molecular structure design is achieved. Therefore, porous carbons controlled at the Ångstrom level are produced by a simple carbonization process of the designed carbon sources. The carbonization yields of these 3D aromatic molecules are quite high; thus these carbon sources can be used as carbon sources instead of phenolic resin and poly(acrylonitrile). Compared with the template carbonization method, this synthetic method does not require inorganic templates. However, by using these 3D aromatic molecules as carbon sources in the template

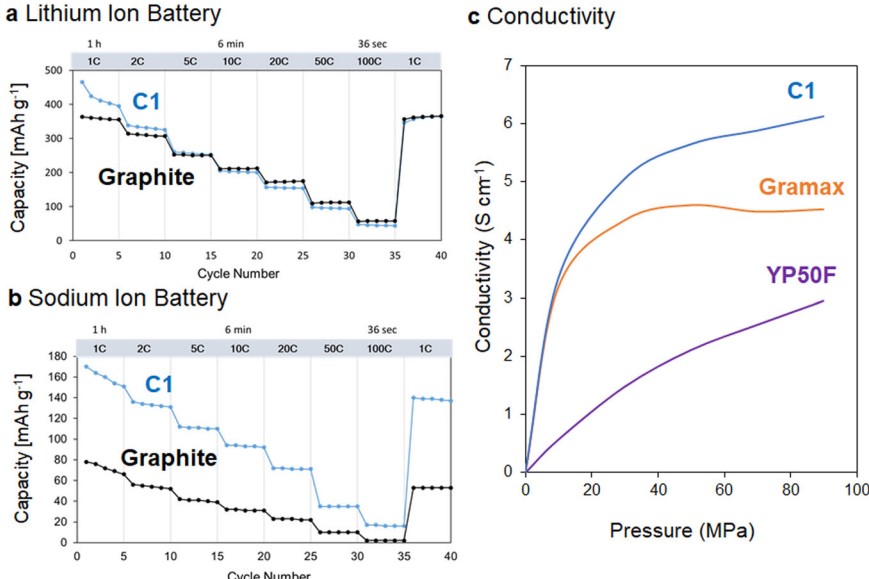

**Fig. 4 Electronic property.** Application of **C1** as an anode active material for **a** lithium- and **b** sodium-ion batteries (1C = 372 mAh g⁻¹). As references, capacities of graphite are also shown. **c** Conductivity of the porous carbon **C1**, Gramax (commercial anode material for a lithium-ion battery) and YP50F (commercial activated carbon for a supercapacitor electrode).

carbonization method, porous carbons with two different pore-size control levels at the Ångstrom and nanometer scales can be produced. We are conducting further investigations to improve on these findings.

## Methods

**Materials**. All solvents and reagents were used as supplied. Tetrakis(4-ethynyl-phenyl)methane **1** was synthesized according to the previous paper[32]. Two reference carbon materials were used for conductivity measurement: YP50F and Gramax. YP50F is a water-vapor-activated carbon made from palm shells and was kindly provided by Kuraray Chemical Co. Ltd. YP50F is used as an electrode material for commercial electrical double-layer capacitors owing to its large surface area (1700 m² g⁻¹) and sufficient conductivity. Gramax is used as an anode material for commercial lithium-ion batteries and was kindly provided by Osaka Gas Chemicals Co. Ltd.

**Synthesis**. Synthetic routes and characterization of the carbon source **2** are shown in Supplementary Figs. 1–4. The synthesis procedures of **2**, MSC-30, and ZTC as well as the LIB and SIB preparation and evaluation, are also shown in Supplementary methods. Carbonization procedure of the carbon sources is shown in Supplementary Fig. 5.

**Measurements**. The ¹H NMR spectra were recorded at 500 MHz with a JEOL-ECA500 spectrometer. TGA was performed using a TG/DTA6200 analyzer, SEIKO Instruments, Inc. under nitrogen atmosphere. The TPD measurement was performed on a sample (ca. 1 mg) by using a home-made high-vacuum apparatus allowing quantitative detection of gas evolutions such as $H_2$, $H_2O$, CO, and $CO_2$, during a heating of the sample up to 1800 °C with a heating rate of 10 °C min⁻¹ [34]. Raman spectroscopy was performed on a JASCO NRS-3300FL spectrometer (laser: 532.2 nm). The electric conductivity of the carbon powders was measured by a two-probe method reported elsewhere[35]. Gas and vapor sorption isotherms were obtained by a BELSORP-max (BEL Japan Inc., Osaka, Japan) sorption analyzer. The surface area was calculated by the BET method. Powder XRD patterns were obtained on Smart Lab (Rigaku) diffractometer with CuKα radiation (tube voltage, 40 kV, tube current, 20 mA). The nanostructures of the samples were directly observed by a transmission electron microscope (JEM-2010, JEOL Ltd.) with an accelerating voltage of 200 kV.

**Molecular probe method**. Adsorption of the various probes was measured at 298 K. Molecular probes of different kinetic diameters, including $CO_2$ (0.33 nm at 298 K), ethane (0.40 nm at 298 K), and *n*-butane (0.43 nm at 298 K) were used for adsorption to evaluate the pore size of the adsorbents[36–38].

## Data availability
The authors declare that all data supporting the findings of this study are available within the Article and its Supplementary Information. The raw data generated in this study are available from the corresponding author upon reasonable request.

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

## Acknowledgements

This work was supported by Grant-in-Aid for Kiban A (JP19H00909, TO) from MEXT Japan, JST CREST (JPMJCR18R3), World Premier International Research Center Initiative (WPI), MEXT, Japan, and "Five-star Alliance" in "NJRC Mater. & Dev.", MEXT Japan.

## Author contributions

T.O. conceived the project and designed the experiments. Y.S., K.O., T.K., and T.Y. synthesized carbon sources. K.T., R.T., and H.N. characterized porous carbon materials. Y.N. and B.D.L.C. used porous carbon materials as anode active materials. All authors analyzed and discussed the results, and co-wrote the paper.

## Competing interests

The authors declare no competing interests.
