## [Peer Review File · Communications Chemistry]

Reviewers' comments:

Reviewer #1 (Remarks to the Author):

In this manuscript, the authors described a template-less preparation of porous carbon materials with pore sizes at the Ångstrom level by proper selection of carbon sources. Tetraphenylmethane and tetrabiphenylmethane with ethynyl groups were used as the carbon sources and high carbonization yields were achieved. The dependence of pore sizes on the starting compounds were observed, and the larger building block, tetrabiphenylmethane, gave a larger pore size of 4.39 Å. The work is a substantial extension of the authors' previous one (Angew. Chem. Int. Eds, 2015, 54, 6466). Due to the important role of the carbon materials in various applications, this reviewer supports its publication. Issues that need to be clarified include:

1. It's better to give the pore size of polymers obtained with these two aromatics with ethynyl groups before calcination.
2. There is no oxygen element in the starting materials, what's the origin of O listed in supplementary Table 1? Is that introduced during carbonization process?
3. Normally, a smaller pore size gives a larger surface area. Why the surface area of C1 is much lower than that of C2 (8.21 vs 451 m²/g) with a pore size difference about 1 Ångstrom?

Reviewer #2 (Remarks to the Author):

The authors report a synthetic method of carbon materials with a pore size that can be precisely controlled at the Ångstrom level by carbonization of the designed carbon sources. The manuscript is well organized, so I would like to ask the authors to revise the following minor points. This work collects excellent data on the target subjects.

How about the materials' stability? This point is very critical if we consider practical applications.

The authors show wide-angle XRD patterns for samples. How about the average crystallite sizes? This size is matched with TEM data?

Related papers have been reported by different research groups. It is better to cite the following refs to support some related paragraphs in the introduction part. Metal-organic frameworks (MOFs) are concerned as potential materials for preparing functional carbons (Angew Chem Int Ed 59, 2066, 2020; Sci Rep 6, 30295, 2016; J. Mater. Chem. A 5, 15356, 2017, Chem, 6, 19-40 (2020), etc.). Recently, various shaped Prussian blue (Bull Chem Soc Jpn, 2019, 92, 875) has been also demonstrated under different conditions. It is better to address these recent works in the introduction part.

Overall the manuscript is well written, but I want to see the authors' perspective (future vision) on this research in the conclusion part.

Reviewer #3 (Remarks to the Author):

This manuscript reports on an attempt to control the pore size of porous carbons prepared via carbonisation by choice of starting material. Two samples are prepared, and the claim is that they

show different pore size determined by nature of the precursor materials. However, this claim is difficult to prove by using only two samples. Furthermore, only one sample, C2, appears to be porous. C1 has very low porosity and, from the presented sorption isotherm, I do not see how the pore size can be determined accurately. The main claim of the manuscript can only be proven by having a much larger series of precursors that provide truly porous carbons with a range of pore sizes. As presented, what we have is simple two different precursors giving two different carbons. This is what would be expected for any precursor.

It is also not clear why there would be a need to prepare such lowly porous carbons via this route. There are many more routes that offer cheaper and more easily carbonised feedstock to similar porosity.

Is control of pore size at Angstrom level unique to this work? In a sense all pore size control can operate at that level especially for carbons derived from aromatic substrates.

Dear Reviewers,

According to three reviewer's comments, we have revised our manuscript. We would like to explain these corrections point-by-point toward each reviewers as follows:

Reviewer #1:

Comment: In this manuscript, the authors described a template-less preparation of porous carbon materials with pore sizes at the angstrom level by proper selection of carbon sources. Tetraphenylmethane and tetrabiphenylmethane with ethynyl groups were used as the carbon sources and high carbonization yields were achieved. The dependence of pore sizes on the starting compounds were observed, and the larger building block, tetrabiphenylmethane, gave a larger pore size of 4.39 Å. The work is a substantial extension of the authors' previous one (Angew. Chem. Int. Ed, 2015, 54, 6466). Due to the important role of the carbon materials in various applications, this reviewer supports its publication. Issues that need to be clarified include:

Response: We appreciate the recommendation of the reviewer to publish this paper.

Comment 1: It's better to give the pore size of polymers obtained with these two aromatics with ethynyl groups before calcination.

Response 1: According to the reviewer's comments, we performed N₂ and CO₂ gas adsorption measurements to evaluate the pore sizes of these aromatics after the polymerization of **1** and **2**. Both polymerized samples adsorbed CO₂ and N₂, thus the porous structures were developed by the thermal polymerization.

We mentioned the point in the revised manuscript as follows:

After the polymerization, both polymerized samples adsorbed CO₂ and N₂ (Supplementary Figs. 7 and 8), thus the porous structures were developed by the thermal polymerization.

Comment 2: There is no oxygen element in the starting materials, what's the origin of O listed in supplementary Table 1? Is that introduced during carbonization process?

Response 2: Generally, carbons after carbonization contain dangling bonds (radicals), which are easily oxidized upon exposure to air. Thus, in these samples, oxygen elements resulted from oxidation of the dangling bonds after the carbonization.

We mentioned the point in the revised manuscript as follows:

The origin of oxygen elements resulted from oxidation of the dangling bonds after the carbonization. Generally, carbons after carbonization contain dangling bonds (radicals), which are easily oxidized upon exposure to air.³¹

Comment 3: Normally, a smaller pore size gives a larger surface area. Why the surface area of **C1** is much lower than that of **C2** (8.21 vs 437 m²/g) with a pore size difference about 1 angstrom?

Response 3: In this study, we used N₂ gas to measure the surface area from BET. Pore size of **C1** is smaller than size of nitrogen molecules (3.6 Å), thus nitrogen molecules cannot access into the inner pore spaces of **C1**. In contrast, the size of **C2** pore is larger than that of nitrogen molecules, thus nitrogen molecules can access into the inner pore spaces of **C2**. Therefore,

surface area of **C1** calculated by nitrogen molecule probes are too small compared with **C2**. We mentioned the point in the revised manuscript as follows:

We also investigated the surface area by using N_2 gas from BET (Supplementary Fig. 11). The surface area of **C1** is much lower than that of **C2** (8.21 vs 437 m^2/g). The size of **C1** pore is smaller than that of nitrogen molecules (3.6 Å), thus N_2 molecules cannot access the inner pore spaces of **C1**. In contrast, the size of **C2** pore is larger than that of N_2 molecules, thus N_2 molecules can access the inner pore spaces of **C2**. Therefore, surface area of **C1** calculated by N_2 molecule probes is too small compared with **C2**.

We also showed the BET data in SI in the revised manuscript.

Reviewer #2:

Comment: The authors report a synthetic method of carbon materials with a pore size that can be precisely controlled at the angstrom level by carbonization of the designed carbon sources. The manuscript is well organized, so I would like to ask the authors to revise the following minor points. This work collects excellent data on the target subjects.

Response: We appreciate the recommendation of the reviewer to publish this paper.

Comment 1: How about the materials' stability? This point is very critical if we consider practical applications.

Response 1: Thank the reviewer to point out the interest. Yes, thermal stability of these porous carbons is very important point for practical applications. In this research, we calcinated these aromatics at 900 °C for 1 h to obtain porous carbons. Therefore, porous carbons **C1** and **C2** were stable by heating again at 900 °C under nitrogen atmosphere. In contrast, typical porous carbons such as ZTC and MSC-30 are not stable after heating these carbons at 900 °C. The surface area of ZTC and MSC-30 were decreased by ca. 40% and 10%, respectively, after heating. We mentioned the point in the text as follows and showed N_2 adsorption measurements (77 K) of 900°C-treated ZTC and MSC-30 in SI.

Thermal stability of these porous carbons is very important point for practical applications. Porous carbon **C2** was stable by heating again at 900 °C under nitrogen atmosphere (Supplementary Fig. 13a). In contrast, typical porous carbons such as ZTC and MSC-30 are not stable at 900 °C. The surface area of ZTC and MSC-30 were decreased by ca. 40% and 10%, respectively, after heating (Supplementary Figs. 13b and 13c).

Comment 2: The authors show wide-angle XRD patterns for samples. How about the average crystallite sizes? This size is matched with TEM data?

Response 2: According to the reviewer's comment, we calculated the average crystalline sizes. The average crystalline sizes calculated by Scherrer formula were ca. 1.8 nm, which are too small to observe by TEM because of no clear scattering contrast to observe these small size carbon crystals into same carbons.

Comment 3: Related papers have been reported by different research groups. It is better to cite the following refs to support some related paragraphs in the introduction part. Metal-organic frameworks (MOFs) are concerned as potential materials for preparing functional

carbons (Angew Chem Int Ed 59, 2066, 2020; Sci Rep 6, 30295, 2016; J. Mater. Chem. A 5, 15356, 2017, Chem, 6, 19-40 (2020), etc.). Recently, various shaped Prussian blue (Bull Chem Soc Jpn, 2019, 92, 875) has been also demonstrated under different conditions. It is better to address these recent works in the introduction part.

Overall the manuscript is well written, but I want to see the authors' perspective (future vision) on this research in the conclusion part.

Response 3: According to the reviewer's comment, we cited these related papers. We cited these papers in refs 23-27. We also added the perspective in the conclusion part as follows:

By using these 3D aromatics as carbon sources in the template carbonization method, porous carbons with two different controlled levels at the angstrom and nanometer scales can be produced. We are conducting further investigations to improve these findings.

Reviewer #3:

Comment 1: This manuscript reports on an attempt to control the pore size of porous carbons prepared via carbonisation by choice of starting material. Two samples are prepared, and the claim is that they show different pore size determined by nature of the precursor materials. However, this claim is difficult to prove by using only two samples. Furthermore, only one sample, **C2**, appears to be porous. **C1** has very low porosity and, from the presented sorption isotherm, I do not see how the pore size can be determined accurately. The main claim of the manuscript can only be proven by having a much larger series of precursors that provide truly porous carbons with a range of pore sizes. As presented, what we have is simple two different precursors giving two different carbons. This is what would be expected for any precursor.

Response 1: We appreciate the comments of the reviewer to improve this paper. To clearly compare the pore sizes of **C1** with **C2**, we improved the evaluation of pore size of **C1**. **C1** took up CO₂ (diameter: 3.3 Å at 298 K), but did not take up N₂ (3.6 Å at 77 K). We plotted micro-pore volumes from CO₂ and N₂ gas probes by DA method (Fig. 2d). Furthermore, according to the reviewer's comment, we prepared porous carbons from another aromatic precursor spirobifluorene (**4**). In this case, we also obtained carbon (**C4**) in high yield. **C4** also had pores, which are similar size of **C1** because acetylene-modified spirobifluorene **4** has three dimensional structure, which is similar to **1**. We added the **C4** data and mentioned the point in the revised manuscript as follows:

We prepared porous carbons from another aromatic precursor spirobifluorene with four acetylene groups (**4**). In this case, the obtained carbon adsorbed CO₂ and ethane (Supplementary Fig. 11), and had pores at 4.07 Å (Supplementary Fig. 12), which are similar size of **C1** because the acetylene-modified spirobifluorene **4** has three dimensional structure, which is similar to **1**. From the result, it is revealed that carbonization of three-dimensional structures with acetylene groups are useful way to prepare porous carbons with controlled pores at angstrom level.

We would like to investigate relationship between building block structures and pore size of the porous carbons using various two- and three-dimensional aromatics with acetylene groups in details, which is our next research. Thank the reviewer point out the interest.

Comment 2: It is also not clear why there would be a need to prepare such lowly porous carbons via this route. There are many more routes that offer cheaper and more easily

carbonized feedstock to similar porosity. Is control of pore size at angstrom level unique to this work? In a sense all pore size control can operate at that level especially for carbons derived from aromatic substrates.

Response 2: As the reviewer mentioned, there are several classical methods to make porous carbons, such as chemical vapor deposition on activated carbons (*Carbon* **36**, 377–382 (1998)). However, in these methods, max carbon yields after the carbonization are ca. 30%, indicating original carbon precursor structures completely destroyed during the activation process. To make porous carbons controlled at micro-pores, extraction of metals from carbides (*Nat. Mater.* **2**, 591–594 (2003)) is well-known. However, the process is costly and generates gases with toxicity and corrosively. Another method is template methods (*Chem. Commun.* **54**, 5648–5673 (2018)), but the templates and removing template from the composites are costly. Furthermore, these methods depend on a craftsmanship process, based on a lot of empirical parameters. Due to the craftsmanship process, it is biggest problems to obtain same porous carbons with controlled pores at angstrom level using procedures reported by different research groups. To produce porous carbons with controlled pores at angstrom level, new methods of simple, versatile and reproducible by anybody have been highly desired. The novelty in this study is preparation of porous carbons resulting from molecular design of the carbon precursors. In this method, we successfully obtained porous carbons controlled at angstrom level by simple carbonization of the carbon precursors. Due to the high carbonization yields, porous carbon structures reflect on the original precursor structures and angstrom level porous size control based on molecular structure design is achieved in this study. In the revised manuscript, we mentioned these points in the text as follows:

Introduction Part:

There are other several classical methods to make porous carbons, such as chemical vapor deposition on activated carbons.¹⁵ However, in these methods, max carbon yields after the carbonization are ca. 30%, indicating original carbon precursor structures completely destroyed during the activation process. To make porous carbons controlled at micro-pores, extraction of metals from carbides is well-known.¹⁶ However, the process is costly and generates gases with toxicity and corrosivity. Furthermore, these methods depend on a craftsmanship process, based on a lot of empirical parameters. Due to the craftsmanship process, it is biggest problems to obtain same porous carbons with controlled pores at angstrom level using procedures reported by different research groups. To produce porous carbons with controlled pores at angstrom level, new methods of simple, versatile and reproducible have been highly desired.

Conclusion Part:

The novelty in this study is preparation of porous carbons resulting from molecular design of the carbon precursors. In this method, we successfully obtained porous carbons controlled at angstrom level by simple carbonization of the carbon precursors. Due to the high carbonization yields, porous carbon structures reflect on the original precursor structures and angstrom level porous size control based on molecular structure design is achieved in this study.

In the previous version, our explanation about these points was insufficient, thus we sorry for the insufficient explanation and thank the reviewer to point out the important points.

Changes made in the manuscript to reviewer's comments are highlighted.

Finally, we wish to thank reviewers for kind and worth comments.

Sincerely yours,

Tomoki

Prof. Dr. Tomoki Ogoshi
Department of Synthetic Chemistry and Biological Chemistry,
Graduate School of Engineering
Kyoto University, Katsura, Nishikyo-ku, Kyoto, 615-8510 Japan
(E-mail) ogoshi@sbchem.kyoto-u.ac.jp
(Phone) +81(Japan)-75-383-2733
(FAX) +81(Japan)-75-383-2732

Reviewers' comments:

Reviewer #1 (Remarks to the Author):

All issues of my previous concerns have been clarified in this version, I believe it can be published as is. The authors' effort on the revision is appreciated.

Reviewer #2 (Remarks to the Author):

The authors have conducted additional thermal stability experiments to answer my query regarding the stability of the prepared porous carbons. Additionally, they have provided perspective regarding the novelty of this study in the Conclusion section. Therefore, I am happy to accept this manuscript in the current version.

Reviewer #3 (Remarks to the Author):

The manuscript has been sufficiently revised and also benefits from the extra experiments and added materials data.

The porosity of one of the samples is still very low and could have been better explored using CO₂ sorption.

The data on stability of activated carbons and ZTC is surprising. Activated carbons and ZTCs are often synthesised at 900 °C and are therefore stable to heating at 900 °C. The authors do not state how the ZTCs were prepared, and therefore the figure in the SI may overall be misleading. More information should be given.

Dear Reviewers,

According to three reviewer's comments, we have revised our manuscript. We would like to explain these corrections point-by-point toward each reviewers as follows:

Reviewer #1:

Comment: All issues of my previous concerns have been clarified in this version, I believe it can be published as is. The authors' effort on the revision is appreciated.

Response: We appreciate the recommendation of the reviewer to publish this paper.

Reviewer #2:

Comment: The authors have conducted additional thermal stability experiments to answer my query regarding the stability of the prepared porous carbons. Additionally, they have provided perspective regarding the novelty of this study in the Conclusion section. Therefore, I am happy to accept this manuscript in the current version.

Response: We appreciate the recommendation of the reviewer to publish this paper.

Reviewer #3:

Comment: The manuscript has been sufficiently revised and also benefits from the extra experiments and added materials data.

Response: We appreciate the recommendation of the reviewer to publish this paper.

Comment: The porosity of one of the samples is still very low and could have been better explored using CO₂ sorption.

Response: Thank the reviewer for pointing out the important point. Yes, the porosity of **C1** is lower than **C2** because size of the carbon source **1** is smaller than **2**. We have explored the porosity of **C1** using CO₂ sorption (Fig. 2b). In addition, the small pore size of **C1** worked better for lithium and sodium ion batteries than that of **C2** (Supplementary Fig. 15). Therefore, the pore size control by molecular design of the carbon sources in this study is very important.

We added the sentence in the revised manuscript as follows:

Therefore, the pore size control by molecular design of the carbon sources in this study is very important.

Comment: The data on stability of activated carbons and ZTC is surprising. Activated carbons and ZTCs are often synthesised at 900 °C and are therefore stable to heating at 900 °C. The authors do not state how the ZTCs were prepared, and therefore the figure in the SI may overall be misleading. More information should be given.

Response: According to the reviewer's comment, we added synthesis procedure of ZTC used in this study in SI. MSC-30 was kindly provided by Kansai Coke and Chemicals Co., Ltd. We mentioned the point in the SI as follows:

MSC-30

MSC-30 was kindly provided by Kansai Coke and Chemicals Co., Ltd.

ZTC Preparation

ZTC was prepared by a traditional two-step method. Well-dried NaY zeolite ($\text{SiO}_2/\text{Al}_2\text{O}_3 = 5.6$, obtained from Tosoh Co., Ltd.) was impregnated with furfuryl alcohol, followed by washing with mesitylene. The zeolite powder accommodating furfuryl alcohol was heated at 150 °C for 8 h under N_2 flow to polymerize the monomer into polyfurfuryl alcohol. The resulting polymer/zeolite composite was heated up to 700 °C in N_2 flow with a heating rate of 5 °C/min. When the temperature becomes 700 °C, the N_2 gas was switched with a mixture of 7%-propylene/ N_2 , and chemical vapour deposition was carried out for 2 h. Then, the gas was switched back to N_2 and a post-treatment was performed at 900 °C for 3 h. After cooling down the sample, zeolite was dissolved away by hydrofluoric acid. Finally, the wet sample was dried at 150 °C for 6 h under vacuum to obtain ZTC.

When heated again at 900 °C under nitrogen atmosphere (Supplementary Fig. 13a), the porous carbon **C2** which was synthesized by template-free simple pyrolysis did not show porosity loss. Such a stable behaviour is comparable to those of conventional activated carbons such as MSC-30 (Supplementary Fig. 13b), and is advantageous over high-performance materials synthesized by template techniques. For example, zeolite-templated carbon (ZTC), an ordered microporous carbon known as a high-performance material, was not stable at 900 °C. The surface area of ZTC was decreased by ca. 40% after heating (Supplementary Fig. 13c). We mentioned points in the revision.

Changes made in the manuscript to reviewer's comments are highlighted.

Finally, we wish to thank reviewers for kind and worth comments.

Sincerely yours,

Tomoki

Prof. Dr. Tomoki Ogoshi
Department of Synthetic Chemistry and Biological Chemistry,
Graduate School of Engineering
Kyoto University, Katsura, Nishikyo-ku, Kyoto, 615-8510 Japan
(E-mail) ogoshi@sbchem.kyoto-u.ac.jp
(Phone) +81(Japan)-75-383-2733
(FAX) +81(Japan)-75-383-2732

REVIEWERS' COMMENTS:

Reviewer #3 (Remarks to the Author):

The manuscript has been sufficiently revised and the question on thermal stability has been addressed.